# Neural Boneprint: Person Identification from Bones using Generative Contrastive Deep Learning

## ABSTRACT

Forensic person identification is of paramount importance in accidents and criminal investigations. Existing methods based on soft tissue or DNA can be unavailable if the body is badly decomposed, white-ossified, or charred. However, bones last a long time. This raises a natural question: *can we learn to identify a person using bone data?* We present a novel feature of bones called *Neural Boneprint* for personal identification. In particular, we exploit the thoracic skeletal data including chest radiographs (CXRs) and computed tomography (CT) images enhanced by the volume rendering technique (VRT) as an example to explore the availability of the neural boneprint. We then represent the neural boneprint as a joint latent embedding of VRT images and CXRs through a bidirectional cross-modality translation and contrastive learning. Preliminary experimental results on real skeletal data demonstrate the effectiveness of the Neural Boneprint for identification. We hope that this approach will provide a promising alternative for challenging forensic cases where conventional methods are limited. The code will be available at ***.

## CCS CONCEPTS

• **Computing methodologies** → **Computer vision tasks**; • **Applied computing** → *Evidence collection, storage and analysis*; Life and medical sciences.

## KEYWORDS

Person Identification, Skeletal Data, VRT, CXR, Cross Modality, Deep Learning

## 1 INTRODUCTION

> *"I am not leaving until these bones lead me to wherever my husband is!"*
> – "Bones", Season 11, Episode 1, FOX 2015

> *The artwork once yearned there was a technique that could find some subtle clues by analyzing the bones of the victim.*

Person identification is the primary and initial concern in accidents and criminal investigations. For a long time, researchers have explored various biological evidence for person identification and authentication, including genetic material deoxyribonucleic acid

*ACM MM, 2024, Melbourne, Australia*
© 2024 Copyright held by the owner/author(s). Publication rights licensed to ACM.
ACM ISBN 978-x-xxxx-xxxx-x/YY/MM
https://doi.org/10.1145/nnnnnnn.nnnnnnn

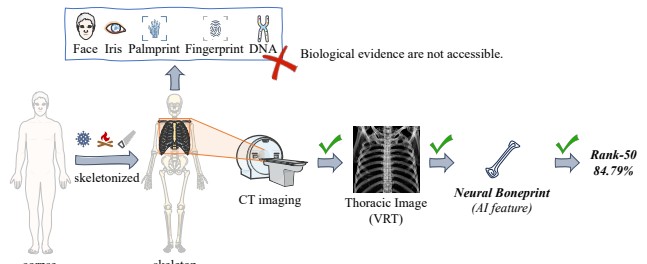

**Figure 1: When the corpse was deeply skeletonized (highly decomposed, charred, or deliberately destroyed), biological evidence involving soft tissue or DNA is not available. We present *Neural Boneprints* for person identification by taking the thoracic skeleton as an example.**

(DNA) [54], faces [9, 14, 45, 57], fingerprints [36, 49], palmprints [28, 47, 61], irises [11, 49, 62], etc.

However, in forensics, when a body/corpse is extensively decomposed, intentionally damaged, or burned, soft tissue markers such as facial features and fingerprints are often neither viable nor extractable for identification. In addition, DNA extraction from such remains is unfortunately extremely challenging due to the progressive degradation of DNA over time [17]. Beyond the technical difficulties, the identification process is constrained by financial costs, time requirements, and the extent of available DNA databases. The efficacy of DNA analysis heavily relies on the presence of the individual's DNA sequence, or that of close relatives, within these databases. Without prior DNA sequencing and storage, identification becomes significantly hindered, illustrating the limitations of current forensic methodologies in certain scenarios. The intractability and accessibility of biological evidence (e.g. DNA) to skeletons hinders modern forensic identification for skeletons.

Fortunately, there is an important but seldom-exploited basic fact that *bones generally persist for a very long time, either in the corpse or in the skeleton.* Forensic studies have shown that manual identification of persons through comparison of some skeletal imaging materials between antemortem and postmortem is practical [2, 13, 18, 21, 42]. However, these analyses generally depend on the experience of forensic experts. This raises a natural question: ***can we learn to identify a person directly from skeletal data?*** To do this, two fundamental questions need to be answered. First, what skeletal data should we use; second, what kind of features should we learn for identification, and how?

In this paper, we demonstrate that thoracic skeletal data, specifically CXRs and VRT images, are useful for learning to identify directly. We then introduce the Neural Boneprint (NBP), a joint latent embedding extracted through a bidirectional cross-modality translation and contrastive deep learning on CXRs and VRT images, complementing traditional biological metrics. Experimental results

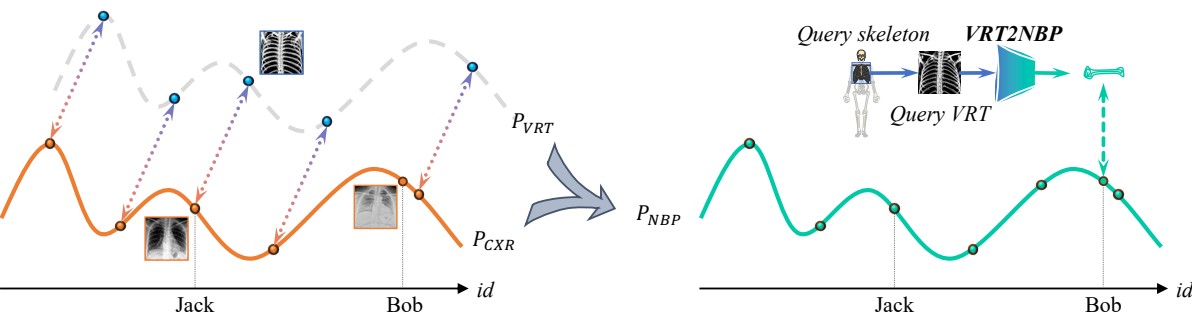

**Figure 2: The VRT images (blue point) are distributed sparsely ($P_{\mathrm{VRT}}$, gray, dashed line) while the CXRs (orange point) are easily collected with a dense distribution ($P_{\mathrm{CXR}}$, orange, solid line). We design a generative contrastive deep learning approach to learning the identity-based non-linear mapping (orange to blue gradient bidirectional arrows) of different modalities. We realize the identifiable complementary interpolation from one modality to another to obtain the unique feature, which is biometric-like but data-driven and named neural boneprint. VRT2NBP (blue to green gradient trapezoid) obtains NBPs from VRT images.**

on real data demonstrate the effectiveness of NBP for person identification in forensics. In summary, the contributions of this work are as follows:

(i) We present a novel perspective on person identification: learning identifiers called *Neural Boneprint* (NBP) from skeletal image data, demonstrating the potential of NBP as a biometric-like identifier that complements traditional forensic methods.

(ii) We present a novel deep learning framework for extracting NBP from CXR and VRT images of the thoracic skeleton. NBP is architecture agnostic, allowing its extraction using different networks.

(iii) Experimental results on real clinical data demonstrate the effectiveness of NBP in identifying people, achieving a Rank-50 identification accuracy of 84.79% - *twice* the performance of related state-of-the-art methods.

## 2 MOTIVATION

In this paper, we aim to explore the important but never exploited basic fact that *bones generally persist for a very long time, either in the corpse or in the skeleton.* We assume that there is an implicit feature *boneprint* in bones, similar to palmprints and fingerprints, which encodes identity information and is widely present in skeletons and skeletal data.

### 2.1 Which Bones? Thoracic Skeleton

It is worth noting that not each bone on the skeleton can be a suitable candidate for large-scale person identification. Those available bones, such as vertebrae and skulls [29, 42], contain identifiable boundaries and rich morphology that varies from person to person. In brief, an efficient skeleton-based person identification requires the skeletal data (i) to contain identifiable boundaries and distinct morphology, and (ii) to be easily collected and organized in a large-scale matching pool.

While it is commonly acknowledged that human faces and skulls contain identifiable features [29], we seek to investigate overlooked

aspects of the skeletal structure that may also hold valuable identity information, and the thoracic skeleton serves as an apt example for this purpose. Thoracic skeletons, including ribs, vertebrae, and sternums [19, 25, 39, 56], have been used as manual comparison materials to estimate sex and age due to complex morphology and distinct visual individual differences. Therefore, in this paper, we will explore the thoracic skeleton data to learn thoracic boneprints.

### 2.2 Which Thoracic Skeletal Data? VRT & CXR

Forensic pathologists are often required to preprocess computed tomography (CT) images by volume rendering technology (VRT) [16, 46] to obtain VRT images for analytical studies. For the corpse, chest VRT can obtain a clear thoracic skeleton to avoid the effects of soft tissue and organ decomposition on skeletal observation. However, it is impractical to construct an adequate pool from VRT images. That is, not everyone has had a CT scan because the risk of radiation-induced cancer is increased by the use of CT [4, 34], resulting in a paucity of VRT images. As a result, it is unlikely that there is a pre-stored CT image to compare with a query VRT image of an unnamed corpse.

Fortunately, the chest X-ray (CXR) is a regular and inexpensive part of the physical examination. In industrialized countries, there is an average acquisition of 238 erect-view CXRs per 1000 of the population annually [5, 35]. In 2006, approximately 129 million CXRs were obtained in the United States alone [33]. The CXR is often the first imaging study acquired and remains central to screening, diagnosis, and management of a broad range of conditions [40, 50]. After the COVID-19 pandemic, that number will only be higher. More importantly, CXR images usually contain all the skeletal elements of the chest and their identities. This shows that CXRs can naturally form a huge pool for the identification of thoracic skeletons. Simply using CXRs or synthetic CXRs of corpses may be proposed. However, for corpses that are not fully decomposed, the decomposition and expansion of organs and soft tissues can significantly impact observation and comparison. Furthermore, for skeletonized corpses, the simulated soft tissues and organs in synthetic CXRs can be misleading and thus unsuitable for analysis. The

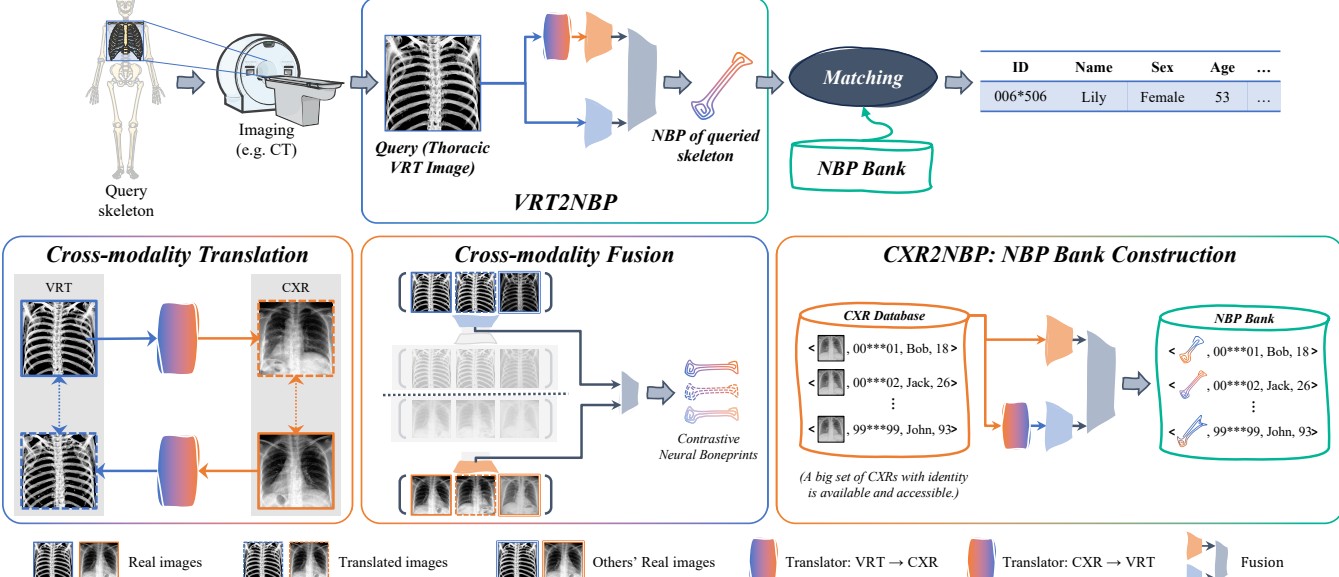

**Figure 3: Top: Query phase. For a query skeletal corpse, we obtain the query** NBP **from the query skeletal data (VRT image) in the** VRT2NBP **process. Then match the query** NBP **with the nearest one in the** NBP **bank. Bottom: Training phase of the cross-modality translation and cross-modality fusion modules, and construction of the** NBP **bank via the** CXR2NBP **process.**

question then becomes, *how can we exploit the vast amount of CXR data and introduce its rich identification information into the small VRT data? If we can learn boneprint from CXR and VRT images and how?*

## 2.3 Comparison Visualization

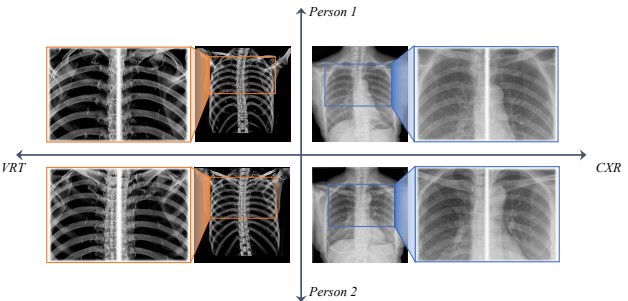

**Figure 4: VRT images and CXRs from two individuals who possess similar overall thoracic skeletal morphology.**

We have visualized VRT images and CXRs from two individuals who possess similar overall thoracic skeletal morphology in Figure 4. Although this similarity might obscure obvious differences at first glance, upon closer inspection, subtle variations in chest contours and rib boundary details become apparent. It is noteworthy that even within skeletons exhibiting overall similarity, these minute distinctions are present. In thoracic skeletons displaying significant morphological differences, the disparities would undoubtedly be more prominent.

**Challenges.** To answer the above questions, several challenges related to the characteristics of skeletal data (VRT, CXR) need to be addressed. The main challenges can be categorized as follows:

(1) *Data Imbalance and 2 Shots.* There is also a notable imbalance in the availability of image types. VRT images are relatively scarce and sparse in distribution, in contrast to CXRs, which are abundant and densely distributed. This disparity presents a significant challenge in data handling and analysis. Besides, we have only 2 shots in one category (a single CXR-VRT image pair per individual) in our dataset. This limitation exacerbates the difficulty in extracting reliable boneprint features.

(2) *Large Intra-Class Variation.* The considerable modality gap between VRT images and CXRs results in significant intra-class variation. CXRs, in particular, exhibit complex overlaps of various anatomical structures, unlike VRT images that predominantly display skeletal features. Additionally, the varied postures during imaging lead to bone deformation, further complicating the analysis.

(3) *Small Inter-Class Difference.* The subtlety of thoracic skeletal differences between individuals poses a formidable challenge. Unlike other identification tasks where inter-class features are distinct, the skeletal features in our case are less conspicuous, sometimes barely discernible to the human eye. This issue is compounded by the physiological similarities in rib numbers and orientations shared among humans, making differentiation based on these features challenging.

Overcoming the above challenges requires sophisticated approaches to image processing, data analysis, and feature extraction to effectively learn and use boneprints for personal identification.

In the next section, we will present a novel deep learning framework to solve these challenges and learn neural boneprints for indentification.

## 3 NEURAL BONEPRINT (NBP)

We aim to use deep learning to learn the boneprints of thoracic skeletons, i.e. neural boneprint (NBP), from VRT images and CXRs for person identification. In particular, to overcome the aforementioned challenges, we propose three modules to learn NBP: (1) Cross-modality Translation. Images are translated into each other's modality to bridge the modality gap and enhance data completeness. (2) Cross-modality Fusion. A dual reconstruction network with contrastive learning fuses fine-grained representations and optimizes inter- and intra-class distances to extract NBPs. (3) NBP-Bank construction. An NBP Bank is constructed from CXR data. Query VRT images are matched against NBPs in the bank for identification.

In this paper, let $x$ and $y$ be the VRT image and CXR image, $\mathsf{T}_{xy} : x \to y$ and $\mathsf{T}_{yx} : y \to x$ are two domain translators. The algorithm ultimately outputs a neural boneprint (NBP) feature through a fusion network F, represented as $\mathsf{F} = \mathrm{NBP}(x, y)$. In the below sections, we will provide a comprehensive description of our NBP algorithm, elucidating its workflow, proposed modules, and design details. The training detail, including the loss function, will also be discussed.

### 3.1 Cross-Modality Translation

The primary obstacle in converting between VRT images and CXRs is the significant differences in anatomical structures and the distortion of the thoracic skeleton [20]. This discrepancy creates a significant gap between the two modalities that cannot be effectively bridged by straightforward unidirectional translation [60] methods like [27].

From a probabilistic standpoint, focusing solely on $p(\mathrm{CXR}|\mathrm{VRT})$ or $p(\mathrm{VRT}|\mathrm{CXR})$ in a unidirectional approach hinders our ability to explore the intricate semantic relationships between VRT images and CXRs. To address this, we propose a bidirectional process, that is, simultaneously learning from VRT to CXRs and vice versa. By deliberately aligning their intermediate latent variables in terms of distribution, we establish a preliminary joint latent space between VRT and CXRs, denoted as $p(\mathrm{VRT}, \mathrm{CXR})$. We argue that this shared latent space enhances the model's capacity to grasp the complex semantic interplay between VRT and CXRs during the cross-modal transformation, as shown in Figure 5. Specifically, we employ two transformation networks to model the processes of $p(\mathrm{CXR}|\mathrm{VRT})$ and $p(\mathrm{VRT}|\mathrm{CXR})$ separately. We consider the latent variables in the networks' middle layer to follow a multivariate Gaussian distribution and compute the 2-Wasserstein distance $W$ to measure the divergence between them:

$$
\begin{aligned}
W_2^2(\mathbf{z}_{\mathrm{VRT}}, \mathbf{z}_{\mathrm{CXR}}) = &\|\boldsymbol{\mu}_{\mathrm{VRT}} - \boldsymbol{\mu}_{\mathrm{CXR}}\|^2 \\
&+ \mathbf{Tr}(\Sigma_{\mathrm{VRT}} + \Sigma_{\mathrm{CXR}} - 2(\Sigma_{\mathrm{VRT}}^{1/2}\Sigma_{\mathrm{CXR}}\Sigma_{\mathrm{VRT}}^{1/2})^{1/2}),
\end{aligned}
\tag{1}
$$

where latent variables are denoted as $\mathbf{z}_{\mathrm{VRT}} \sim \mathcal{N}(\boldsymbol{\mu}_{\mathrm{VRT}}, \Sigma_{\mathrm{VRT}})$ and $\mathbf{z}_{\mathrm{CXR}} \sim \mathcal{N}(\boldsymbol{\mu}_{\mathrm{CXR}}, \Sigma_{\mathrm{CXR}})$. Two transformation networks are $\mathsf{T}_{xy} : x \to y$ and $\mathsf{T}_{yx} : y \to x$.

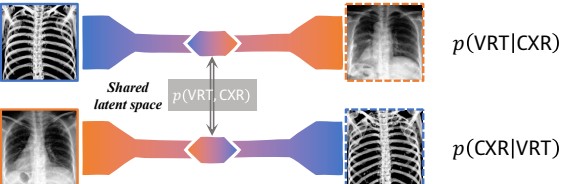

$p(\mathrm{VRT}|\mathrm{CXR})$

$p(\mathrm{CXR}|\mathrm{VRT})$

**Figure 5: Model the processes of $p(\mathrm{CXR}|\mathrm{VRT})$ and $p(\mathrm{VRT}|\mathrm{CXR})$ to obtain the shared latent space.**

In addition, we adopt an extra $\ell_2$ loss penalty to maintain fine-grained personally identifiable information. That is, the real VRT image is paired by identity with the CXR image at the semantic level, so the translated VRT image should be the same as the real VRT image due to the individual identity. In this manner, translated images can be viewed as sampling in the vicinity of the true image in manifold space [58].

### 3.2 Cross-Modality Fusion

After the cross-modality translation, we obtain a real and translated image pair for each individual. Due to the identifiable design, the translated distribution is already close enough to the real distribution in the identity manifold, so we also treat them as real.

We employ a dual input reconstruction network based on contrastive learning to fuse distinguishable skeletal representations to a latent embedding F named neural boneprint. It maps each VRT-CXR image pair to a joint latent embedding in manifold space where the identity information is the primary constraint [14, 31]. It contains a VRT encoder-decoder module, a CXR encoder-decoder module, and a latent fusion module. Each encoder-decoder module is utilized to reconstruct real or translated images for learning the latent fine-grained skeletal representations and then fusing them as neural boneprints. We compute the mean squared error (MSE) between the reconstructed and original images in the pixel space to assist in learning the better fine-grained identical latent features. We also compute the contrastive loss [3, 10, 22] based on the fused embeddings of the real, translated, and the other's real image pairs for minimizing the intra-class distance and maximizing the inter-class distance. During the training phase, we jointly train encoder-decoder modules and the latent fusion module. In the application phase, we ignore decoders and only employ encoders and the latent fusion module with weights frozen.

### 3.3 NBP-Bank Constrtuction

The main idea is that CXRs are widely available and identity-rich, while VRT images are common for skeletal remains. This process aims to create an identification function $\mathrm{ID}(x)$ for efficient identification using a CXR database. Given a CXRs database, we process as following 3 steps:

(1) *Modality Unification:* All CXRs images $\{y_i\}$ are first translated into VRT images $\{\mathsf{T}_{yx}(y_i)\}$ using a translator $\mathsf{T}_{yx}$.
(2) *Joint Embedding Generation:* CXR-VRT pairs $\{(\mathsf{T}_{yx}(y), y)\}$ are mapped via F to joint latent embeddings (NBPs) that capture cross-modal relationships, incorporating identity information.

(3) NBP-*Bank Creation:* The NBP-Bank, denoted as C, is constructed as a searchable table with embeddings as keys and identities as values:

$$C \triangleq \{(\text{Key} = F(T_{yx}(y_i), y_i), \text{Value} = \text{ID}(y_i))\} \quad (2)$$

### 3.4 Query

As we presented above, the modality unification bridges the CXR and VRT domains, improving the accuracy of identification. The joint embeddings (NBP) can effectively represent cross-modal relationships and identity information. Finally, the constructed NBP bank will enable identification via matching, i.e. table lookup, facilitating efficient matching of new VRT images with identities in the CXR database. In particular, given a query VRT image $x$, we proceed as follows 3 steps:

(1) *Modality Translation*: Translate $x$ into its corresponding CXR representation, denoted as $T_{xy}(x)$.
(2) *Joint Embedding Generation*: Map the image pair $(x, T_{xy}(x))$ to a joint latent embedding, i.e., $\text{NBP} = F(x_i, T_{xy}(x_i))$.
(3) *Identification via Nearest Neighbor Search*: Identify the individual associated with $x$ by performing a nearest neighbor search in the reference NBP bank C. This involves determining the NBP $\forall c_i \in C$ with the minimum distance $d$ to $c$, i.e.

$$\text{ID}(x) = \arg\min_i d(c, c_i) \quad (3)$$

## 4 EVALUATION

### 4.1 Datasets Setup

All images used in this work are real clinical data collected from ** Hospital. We construct two datasets and the first contains VRT-CXR image pairs collected from 1,315 healthy individuals without pulmonary diseases or thoracic skeletal lesions. We randomly split the first dataset into training and test sets. The training set contains VRT-CXR image pairs from 1,052 individuals, representing 80%. The test set contains the others, a total of 263 VRT-CXR image pairs, accounting for 20%. The second dataset contains CXRs collected from 874 healthy individuals. The code and demos will be available at ***.

### 4.2 Metrics

**Rank-$k$.** As a classical metric in face identification and person re-identification [52, 59], Rank-$k$ is suitable for our single-shot person identification. In addition, person identification generally involves ethical or legal issues in forensic science. It is also important to simplify the list of candidate victims [1]. Therefore, we propose to use the Percentile Ranking Rate as the metric.

**Percentile Ranking Rate.** The percentile ranking rate aims to measure the ability of the model to hit targets in the top $p$ percent across databases of different scales. Similar to the Rank-$k$ identification rate, we will compute the similarities between the query and the candidates one by one. Then sort the similarities from largest to smallest. If the correct pair is in the top $p$ percent, the match is considered successful, otherwise, the match fails. The ratio of successful pairs to all pairs is the percentile ranking rate.

### 4.3 Qualitative Analysis

The VRT images in the test set of the first dataset are used as queries. We apply our approach to two NBP banks, which are organized from our dataset. One is extracted from the CXRs of the first dataset. To test robustness, we introduce the CXRs in the second dataset as distractors and contribute to the second NBP bank with the test set of the first dataset together. More experiments under large-scale databases are prepared in Supplementary.

**Alternative Comparison.** Since this is the first comprehensive work in this new task, there is no comparable thoracic skeleton identification network. We employ several classical identification models for comparison. In Table 1, we compare the performance of Ours and other alternatives in extracting neural skeletal features. The results of IResNet-18 and IResNet-50 [15] show that deeper networks did not lead to significant gains. We also compared with a classical cross-modality person re-identification method [38].

**Cumulative Match Characteristic.** We select the six methods with the best performance from all the experiments and plot the cumulative match characteristic curve [52], as shown in Figure 6.

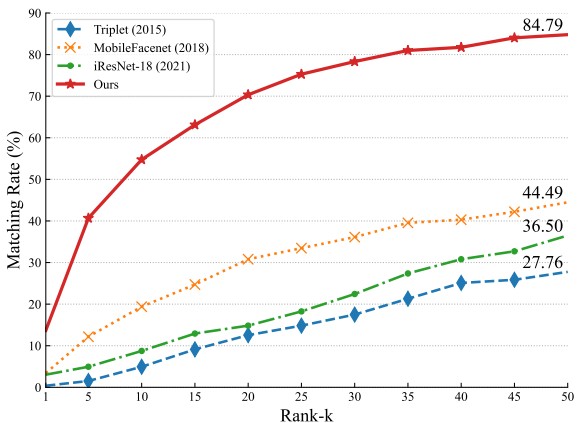

**Figure 6: Cumulative match characteristic curve of Rank-$k$ for different strategies to learn NBP. We achieved state-of-the-art results along different $k$ and obtained at least twice the performance of related approaches.**

**Ablation Study.** We explore the ablation study on the extent of the introduction of CMT in Table 2. It shows that only using one modality (whether CXR or VRT) cannot be effective. We also explore the ablation study of the positive sample strategies for contrastive learning in the neural boneprint extraction step, as shown in Table 3. The Augmented one uses the augmentation strategy of SimCLR [10] without cross-modality translation results introduced. Besides, we explore the ablation study on the choice of the reconstruction loss functions with the CMT and the contrastive loss fixed, as shown in Table 4. The suboptimal result means the L1 loss can capture fine-grained details than MSE in some specific samples, but MSE can get a better generalization performance for total samples.

**Related work Comparison** We compared ours with some studies [12, 26] based on CT images and CXRs for person identification

| Algorithms | Rank-$k$ Rate (%) ↑ | | | | | | | | Percentile Rank Rate ($p$%) ↑ | | |
|---|---|---|---|---|---|---|---|---|---|---|---|
| | |NBP-Bank|=263 | | | | |NBP-Bank|=1137 | | | | | | |
| | $k=1$ | $k=10$ | $k=20$ | $k=50$ | $k=1$ | $k=10$ | $k=20$ | $k=50$ | $p=1$ | $p=5$ | $p=10$ |
| Triplet [24] | 0.38 | 4.94 | 12.55 | 27.76 | 0.00 | 1.90 | 4.94 | 7.98 | 3.04 | 9.51 | 12.55 |
| MobileFaceNet [9] | 3.42 | 19.39 | 30.80 | 44.49 | 1.14 | 7.60 | 12.17 | 23.19 | 8.75 | 23.95 | 33.46 |
| IResNet-18 [15] | 3.04 | 8.75 | 14.83 | 36.50 | 0.00 | 3.42 | 5.32 | 11.03 | 4.56 | 12.17 | 19.01 |
| IResNet-50 [15] | 0.76 | 8.75 | 12.93 | 28.90 | 0.76 | 5.70 | 8.37 | 12.93 | 6.46 | 13.69 | 23.19 |
| LbA [38] | 0.76 | 4.94 | 8.75 | 24.33 | 0.76 | 4.18 | 8.75 | 23.57 | 5.32 | 29.28 | 49.05 |
| **Ours** | **13.69** | **54.75** | **70.34** | **84.79** | **6.46** | **31.18** | **44.87** | **61.98** | **36.88** | **64.64** | **75.29** |

**Table 1: The ultimate performance of Ours and other alternatives on learning neural features of bones for identification. The queries are VRT images and the matching pool is NBP bank constructed from the CXR database. A higher value is better.**

| Algorithms | Rank-$k$ Rate (%) ↑ | | | | | | | | Percentile Rank Rate ($p$%) ↑ | | |
|---|---|---|---|---|---|---|---|---|---|---|---|
| | |NBP-Bank|=263 | | | | |NBP-Bank|=1137 | | | | | | |
| | $k=1$ | $k=10$ | $k=20$ | $k=50$ | $k=1$ | $k=10$ | $k=20$ | $k=50$ | $p=1$ | $p=5$ | $p=10$ |
| Triplet [24] + CMT. (VRT) | 5.70 | 27.76 | 39.54 | 62.36 | 4.56 | 13.31 | 21.67 | 33.08 | 15.59 | 34.60 | 44.49 |
| Triplet [24] + CMT. (CXR) | 2.66 | 16.73 | 28.90 | 52.85 | 0.76 | 3.04 | 4.56 | 12.17 | 3.42 | 13.31 | 23.19 |
| MobileFaceNet [9] + CMT. | 11.79 | 38.78 | 52.85 | 68.06 | 8.37 | 25.86 | 34.22 | 47.53 | 28.90 | 51.71 | 59.70 |
| IResNet-18 [15] + CMT. | 3.80 | 14.45 | 20.15 | 35.74 | 0.38 | 6.84 | 9.13 | 16.35 | 7.60 | 19.01 | 28.52 |
| IResNet-50 [15] + CMT. | 2.66 | 10.27 | 13.31 | 27.38 | 1.14 | 7.22 | 11.03 | 17.87 | 7.98 | 20.53 | 29.28 |
| **CMT + CMF (Ours)** | **13.69** | **54.75** | **70.34** | **84.79** | **6.46** | **31.18** | **44.87** | **61.98** | **36.88** | **64.64** | **75.29** |

**Table 2: Ablation study on introducing the CMT step (CXR only, VRT only, or both). A higher value is better.**

| Positive samples | Rank-$k$ Rate (%) ↑ | | | | | | | | Percentile Rank Rate ($p$%) ↑ | | |
|---|---|---|---|---|---|---|---|---|---|---|---|
| | |NBP-Bank|=263 | | | | |NBP-Bank|=1137 | | | | | | |
| | $k=1$ | $k=10$ | $k=20$ | $k=50$ | $k=1$ | $k=10$ | $k=20$ | $k=50$ | $p=1$ | $p=5$ | $p=10$ |
| Real CXR, translated VRT image from real CXR | 0.00 | 5.70 | 9.89 | 26.24 | 0.76 | 3.80 | 7.98 | 14.83 | 5.32 | 17.87 | 24.71 |
| Augmented [10] (Real CXR, Real VRT image) | 6.08 | 28.90 | 44.11 | 62.74 | 4.18 | 20.15 | 28.14 | 42.97 | 22.43 | 44.87 | 58.56 |
| **Translated CXR from real VRT image, translated VRT image from real CXR** | **13.69** | **54.75** | **70.34** | **84.79** | **6.46** | **31.18** | **44.87** | **61.98** | **36.88** | **64.64** | **75.29** |

**Table 3: Ablation study on Strategy of determining the positive samples for contrastive learning in the CMF step. The augmented in the second line represents introducing the augmentation strategy of SimCLR [10].**

| Algorithms | Rank-$k$ Rate (%) ↑ | | | | | | Percentile Rank Rate ($p$%) ↑ | |
|---|---|---|---|---|---|---|---|---|
| | |NBP-Bank|=263 | | | |NBP-Bank|=1137 | | | |
| | $k=1$ | $k=20$ | $k=50$ | $k=10$ | $k=20$ | $k=50$ | $p=1$ | $p=10$ |
| L1 | 11.03 | 67.30 | 82.89 | 25.10 | 39.54 | 57.41 | 30.04 | 71.86 |
| MSE | 13.69 | 70.34 | 84.79 | 31.18 | 44.87 | 61.98 | 36.88 | 75.29 |

**Table 4: Ablation study on Loss function for Reconstruction (Cross-modality and Contrastive loss are fixed).**

| Method | |Query| | |Bank| | Performance (PRR, $p$%) |
|---|---|---|---|
| Steady MFV, BoW [26] | 27 | 27 | $p=37.04$, 63.00% |
| CLAHE, DFT, Euclidean [12] | 27 | 27 | $p=55.56$, 74.07% |
| Ours | 263 | 1137 | $p=10$, 75.29% |

**Table 5: Our method was experimented with in a larger searchable bank and demonstrated superior performance. A lower $p$ with a higher accuracy is better.**

in Table 5. Our reimplement did not achieve the performance reported in the original work, to ensure a fair comparison, we directly referenced the origin. It shows that our approach contains a larger searchable bank and obtains better performance.

## 4.4 Quantitative Analysis

We visualize the top 5 samples and the last 5 samples of the identification results after introducing distractors, as shown in Figure 7.

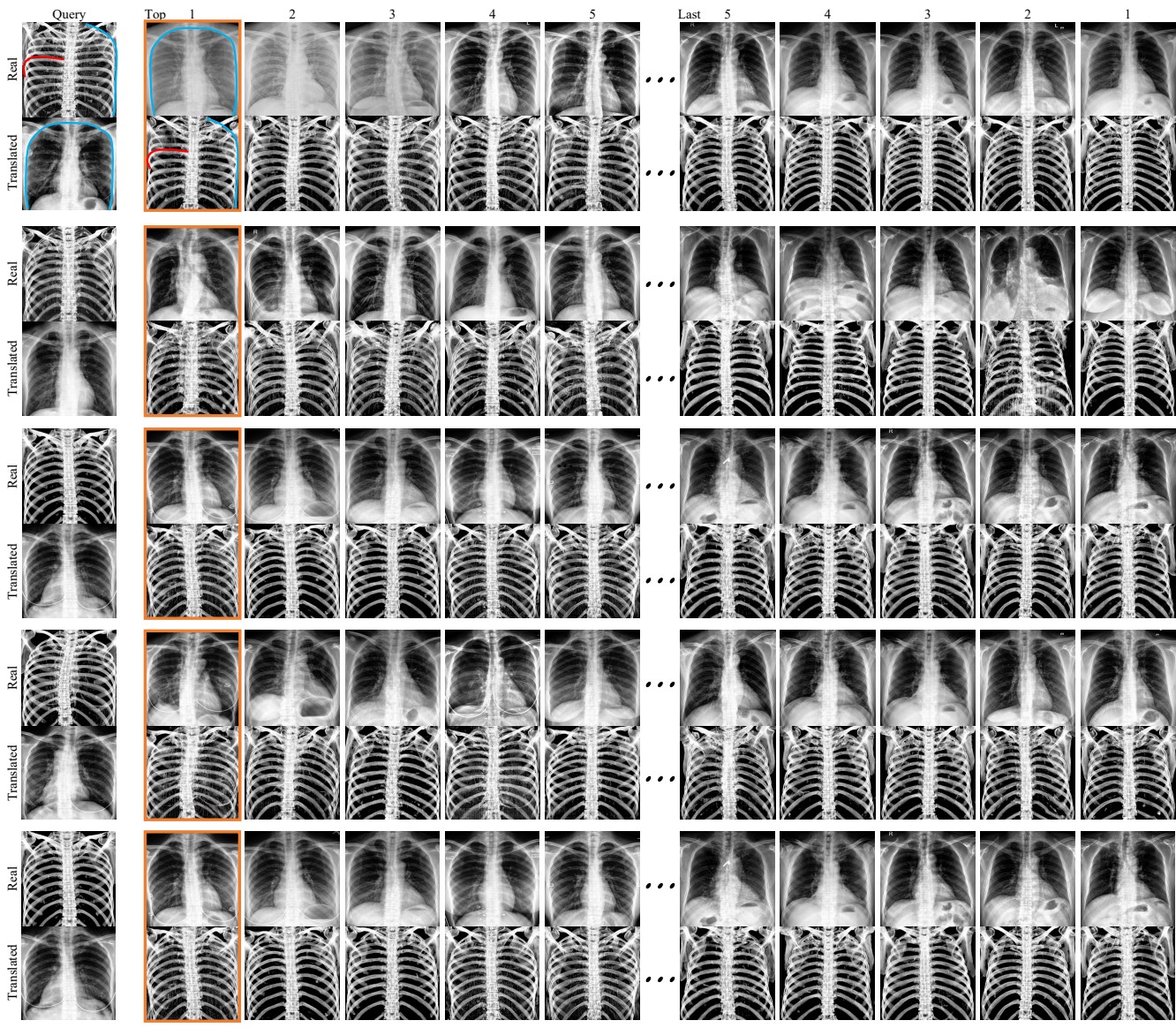

**Figure 7: Visualization of robust identification. The query pair consists of the real VRT image and its translated CXR. The candidate pairs consist of real CXRs with their translated VRT images. The orange rectangle represents the ground truth. The red and blue lines describe the rib boundary and the overall skeletal morphology respectively. The top-5 identification results are similar to the query one while the last are not. More visualization results can be found in Supplementary.**

With careful comparisons, it can be found that the top five image pairs are highly similar to the query image pairs while the last 5 have low similarity. What's more, the deformations of thoracic skeletons and complex overlaps between VRT images and CXRs make it extremely hard for humans to identify, but it can be done by the proposed approach.

Some unidirectional [60] translation methods, such as Pix2Pix [27], may be discussed as alternatives in our cross-modality translation step. With that in mind, we also visualize some VRT-CXR image pairs with their translated results by different methods, as shown in

Figure 8. VRT images only contain thoracic skeletons while CXRs also contain other anatomical structures besides them. Hence, translating CXRs into VRT images is easier than from VRT images to CXRs. We first compare the skeletal visualization of translated VRT images. The unidirectional results show misaligned bones while the cycle results contain distinct morphology with clear boundaries. An important factor is the different postures while taking the VRT images and the CXRs, which leads to the thoracic skeletons being structurally deformed and having more semantic-level mapping

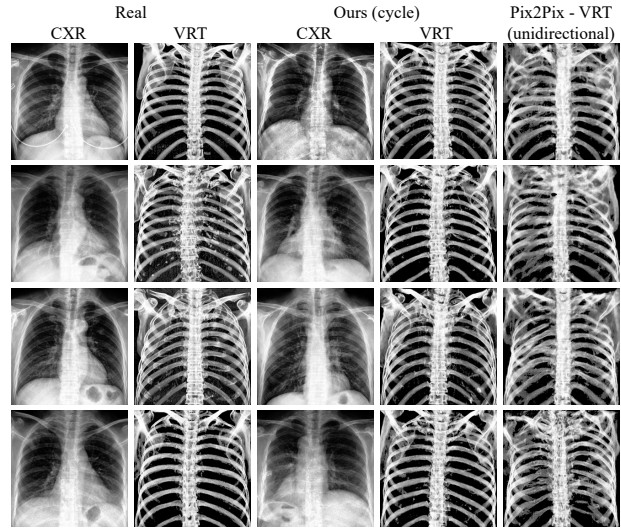

**Figure 8: Cycle translation performs better to solve our task. The translated ones by cycle method are highly similar to the real, both in the overall thoracic skeletal structure and in the boundaries and curvature of the ribs.**

rather than strictly pixel-level. Besides, the corresponding translated VRT images contain almost the same identifiable rib details and orientations compared with the real. Then we compare the visual performance of generating CXRs. As real VRT images do not contain those anatomical structures, some translated structures similar to organs and muscles may be created during the translation. Despite that, we can ignore the created structures because we only focus on the thoracic skeletons.

## 5 RELATED WORK

Identification of human remains through antemortem and postmortem imaging materials [2] has been widely used.

**Manual Methods.** For a long time, forensic odontologists have manually utilized dental radiographs taken of the victim before death and manually compared them to dental data from the remains to assist in the identification [43, 53]. Using head and neck CT and MR imaging to obtain identifying information, including dental findings, to identify individuals is also feasible [18]. Besides, it has been demonstrated that the identification of human remains using visual comparison could be performed by a forensic pathologist with CT experience through 20 cases with antemortem and postmortem CT images, but there is no consensus on the specific number of concordant traits required to establish an identity [2]. Study [51] investigated antemortem and postmortem radiographs of the claviculae and C3-T4 vertebrae to identify skeletons of missing U.S. soldiers from past military operations. The two-dimensional fusion of postmortem computed tomography and antemortem chest radiography makes human identification possible [48]. In summary, these methods not only rely on manual analysis by experienced forensic experts, but are also difficult to deploy on a large scale.

**Automatic Methods.** Some automatic identification methods have been proposed recently. Study [26] utilized the CXR before

death database to match with CT scan image after death. It obtained an accuracy of 63% within 27 subjects and an average ranking of 10 (total 27, 37.04%) based on the extraction and matching of two types of features, the Bag of Words (BoW) and the Histogram of Gradients (HOG). Study [12] utilized morphological erosion to extract rib boundaries and employed Discrete Fourier Transform (DFT) to extract features, which led to an accuracy of 74.07% within rank 15 of 27 subjects. However, the dataset scale of these works is very limited. The performance is not satisfactory and not convincing enough. Study [30] realizes the attention localization and alignment of teeth for person identification by semantic segmentation and creating an atlas with landmarks. However, the annotations of the segmentation and the landmarks require manual creation, which is costly both in time and labor. Besides, this method relies on the spatial arrangement of teeth, while skeletonized corpses usually face loss of teeth. Recently, there has been a burgeoning interest in re-identifying CXRs. Some studies [32, 37, 55] achieve patient re-identification through CXRs for proper archival purposes. Identifying solely through CXRs appears to be more straightforward for our task. However, due to the presence of real internal organs in CXRs, these studies are insufficient to demonstrate the capability for skeletal matching. For those not fully decomposed corpses, the decomposition and expansion of organs can greatly affect comparison. For those skeletonized corpses, the simulated soft tissues can be misleading.

## 6 DISCUSSION

We have shown how machine learning techniques can be used to extract neural boneprint from skeletal information that can be used for identification. We hope this will open up new avenues for identity verification and shed new light on the development and cross-fertilization of different disciplines, including forensics and machine learning. There are many open questions that we and the community need to work on together, and the quality of Neural Boneprint features can be improved from many perspectives. First, as Neural Boneprint features are extracted from VRT and CXR images, the quality of bone detection and imaging is particularly important, and we will investigate more effective imaging methods [6–8] to improve image quality. Second, although the NBPs extracted from VRT and CXR images in this paper can already be used for identity verification, a natural idea is to use more data modalities (e.g., MRI and PET, etc.) to extract NBPs that contain more information. Third, the cross-modality translation could also be a diffusion-style [23, 44] approach. The backbone of cross-modality fusion is also various. The localization of thoracic skeletons in original VRT images and CXRs could be deep learning approaches, such as the YOLO [41] family. Ultimately, we must prioritize stringent privacy protection and ethical review. Hospital data frequently contain highly sensitive personal information and health records of patients. Therefore, when collecting and utilizing such data, it is imperative to adhere strictly to applicable laws and regulations, ensuring comprehensive protection of patients' privacy rights and interests. To sum up, we start an initial discussion on the proposed task and look forward to more works on social equity and judicial justice based on artificial intelligence.

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
