# OpenReview forum: "Neural Boneprint: Person Identification from Bones using Generative Contrastive Deep Learning"
_acmmm.org/ACMMM/2024/Conference — MM2024 Oral_

### Official Review · Reviewer_FWCo · 2024-05-20

**Rating:** 5
**Confidence:** 3

**Summary:**

This paper has proposed the first method for medical imaging-based ReID using rib bones. To fully adopt widely available X-ray images, the authors use the modality translation technique.

**Strengths:**

1. The use of bones for ReID is novel and interesting.
2. The idea of adopting modality translation to use widely available X-ray images as reference datasets is reasonable and interesting, especially considering the impact of the COVID-19 pandemic.
3. The experimental performance seems to be promising.

**Limitations:**

1. How would ReID be affected if parts of the rib bones were fractured, broken, or missing? The authors could design experiments where some bone areas in the query images are randomly masked to testify this. This would effectively validate the method’s robustness in more real-world applications, considering that bodies might have suffered violent trauma before death.
2. What impact would different backbone networks have? Since the authors are the first to work on bone-based ReID, I hope they can test various widespread baselines. This could benefit the community.
4. The technical contribution seems somewhat insufficient. Considering the prior knowledge of rib morphology might be a better approach.
5. Add common ReIDmetrics such as mAP, CMC, INP, etc.
6. In Table 1, have the comparison methods all used your proposed translator? If they perform direct cross-modality retrieval, it seems to be not fair. Please supplement with comparisons where they also use the translator.
7. In Table 5, show both the reproduced and claimed results, and mark them differently for distinction.

**Suitability:**

3

---

### Official Review · Reviewer_Ra9j · 2024-05-21

**Rating:** 5
**Confidence:** 4

**Summary:**

This paper focuses on developing a novel method for forensic person identification using skeletal data. It introduces the concept of Neural Boneprint, a biometric-like identifier extracted from skeletal images. Specifically, it focuses on the thoracic skeleton, using chest radiographs (CXRs) and volume rendering technique (VRT) images. It is an interesting work that might have a wide range of effects.

**Strengths:**

It is an interesting and novel work that might have a wide range of effects.

The motivation is clear , and the proposed includes cross-modality translation and contrastive learning is theoretically sound and technically correct.

Experimental results on real clinical data demonstrate the effectiveness of NBP in identifying people, achieving a Rank-50 identification accuracy of 84.79%.

**Limitations:**

It is an interesting and novel work, I am quite concerned about the robustness of this job. Will abnormal bone morphology affect the identification results? Additionally, considering that metals in clothing will affect X-ray imaging, will this affect the identification effect?

Could authors provide some conclusions about interpretability, such as the model's Region of Interest(RoI).

The reviewer is much concerned about the fairness of the model, especially for different ages and genders.

**Suitability:**

2

---

### Official Review · Reviewer_HVso · 2024-05-24

**Rating:** 4
**Confidence:** 3

**Summary:**

The author of this article proposes the use of the Neural Boneprint (NBP) to address person identification challenges. The experimental process, designed by the author, is grounded in task scenarios and real-world conditions. Specifically, a deep learning framework is proposed to extract NBP from chest X-rays (CXR) and volume rendering technology (VRT). This framework employs techniques such as Cross-modality Translation and Cross-modality Fusion to mitigate issues related to data imbalance, small intra-class variation, and large inter-class differences. Moreover, CXR is utilized to construct an NBP Bank. Experimental results demonstrate that this method achieves outstanding performance, offering novel insights for the development and integration of different disciplines, including forensic science and machine learning.

**Strengths:**

In this paper, a deep learning framework is employed to effectively address challenges such as data imbalance, within-class variation, and between-class differences, thereby enabling the learning of Neural Boneprint (NBP) for person recognition. Experimental results based on real clinical data demonstrate the efficacy of NBP in person recognition, achieving a Rank-50 recognition accuracy of 84.79%, which is twice the performance of the latest existing methods.

1. The experimental design of the study is commendable, as it leverages a substantial amount of CXR data to create a recognition bank and utilizes VRT to design task scenarios for identity recognition in forensic contexts.
2. There exists a significant modal difference between CXR and VRT, coupled with data imbalance issues. The framework proposed in the paper employs Cross-modality Translation and a bidirectional process to address the complex semantic relationship issues between these modalities.
3. Multi-modal fusion is utilized, incorporating “Others’ Real images,” “Translated images,” and “Real images” to tackle the problems of large intra-class variation and small inter-class differences.

Overall, the proposed framework demonstrates significant advancements in person recognition, offering promising avenues for future research in the intersection of forensic science and machine learning.

**Limitations:**

1. The paper mentions, "we believe that this shared latent space enhances the ability of the model to capture the complex semantic interaction between VRT and CXR during cross-modal transformation," but this claim lacks empirical support or detailed elaboration.

2. The paper does not provide an in-depth explanation of the method used to address intra-class differences. Given the considerable modal differences between VRT and CXR images, it is crucial to offer a detailed account of how the model manages these discrepancies.

3. The dataset comprises only healthy individuals and does not consider the potential partial loss of NBP information post-mortem. Consequently, the accuracy and robustness of NBP for forensic identification have not been fully evaluated or validated.

### Suggestions for Improvement

1. Provide a more detailed description of the implementation details of cross-modal transformation and fusion techniques.
2. Conduct research and experiments using more diverse datasets.
3. Include additional performance metrics for a more comprehensive evaluation of the model.
4. In Figure 3, the identifier for “Others’ Real images” is too similar to that for Real images.
5. The description in section “4.2 Metrics” only explains the rationale for choosing Rank-K and Percentile Ranking Rate. Providing additional descriptions would help readers better understand these metrics.

**Suitability:**

2

---

### Meta-Review · Area_Chair_sRZ4 · 2024-06-29

**Recommendation:** Accept (Oral)
**Confidence:** 5

**Metareview:**

All three reviewers are positive about the submission, which addresses the challenging problem of human identification from bones. Some concerns were raised but mostly resolved by the rebuttal, and the final recommendations remain unanimously positive. We agree with the reviewers on the merit of the work and hence recommend its acceptance to ACMMM.